# Real-Life Therapeutic Concentration Monitoring of Long-Acting Cabotegravir and Rilpivirine: Preliminary Results of an Ongoing Prospective Observational Study in Switzerland

**DOI:** 10.3390/pharmaceutics14081588

**Published:** 2022-07-29

**Authors:** Paul Thoueille, Susana Alves Saldanha, Fabian Schaller, Aline Munting, Matthias Cavassini, Dominique Braun, Huldrych F. Günthard, Katharina Kusejko, Bernard Surial, Hansjakob Furrer, Andri Rauch, Pilar Ustero, Alexandra Calmy, Marcel Stoeckle, Manuel Battegay, Catia Marzolini, Pascal Andre, Monia Guidi, Thierry Buclin, Laurent A. Decosterd

**Affiliations:** 1Service and Laboratory of Clinical Pharmacology, Department of Laboratory Medicine and Pathology, Lausanne University Hospital and University of Lausanne, 1011 Lausanne, Switzerland; paul.thoueille@chuv.ch (P.T.); susana.alves-saldanha@chuv.ch (S.A.S.); fabian.schaller@chuv.ch (F.S.); pascal.andre@chuv.ch (P.A.); monia.guidi@chuv.ch (M.G.); thierry.buclin@chuv.ch (T.B.); 2Service of Infectious Diseases, Department of Medicine, Lausanne University Hospital and University of Lausanne, 1011 Lausanne, Switzerland; aline.munting@chuv.ch (A.M.); matthias.cavassini@chuv.ch (M.C.); 3Department of Infectious Diseases and Hospital Epidemiology, University Hospital Zurich, 8091 Zurich, Switzerland; dominique.braun@usz.ch (D.B.); huldrych.guenthard@usz.ch (H.F.G.); katharina.kusejko@usz.ch (K.K.); 4Institute of Medical Virology, University of Zurich, 8057 Zurich, Switzerland; 5Department of Infectious Diseases, Inselspital, Bern University Hospital, University of Bern, 3010 Bern, Switzerland; bernard.surial@insel.ch (B.S.); hansjakob.furrer@insel.ch (H.F.); andri.rauch@insel.ch (A.R.); 6Division of Infectious Diseases, Faculty of Medicine, Geneva University Hospitals, 1205 Geneva, Switzerland; pilar.usteroalonso@hcuge.ch (P.U.); alexandra.calmy@hcuge.ch (A.C.); 7Department of Medicine, Faculty of Medicine, University of Geneva, 1205 Geneva, Switzerland; 8Division of Infectious Diseases and Hospital Epidemiology, University Hospital Basel, University of Basel, 4031 Basel, Switzerland; marcel.stoeckle@usb.ch (M.S.); manuel.battegay@usb.ch (M.B.); catia.marzolini@usb.ch (C.M.); 9Faculty of Medicine, University of Basel, 4031 Basel, Switzerland; 10Department of Molecular and Clinical Pharmacology, Institute of Translational Medicine, University of Liverpool, Liverpool L69 3GF, UK; 11Centre for Research and Innovation in Clinical Pharmaceutical Sciences, Lausanne University Hospital and University of Lausanne, 1011 Lausanne, Switzerland; 12Institute of Pharmaceutical Sciences of Western Switzerland, University of Geneva, 1206 Geneva, Switzerland

**Keywords:** long-acting antiretroviral therapy, cabotegravir, rilpivirine, therapeutic drug monitoring, pharmacokinetics, population pharmacokinetic modeling, pharmacokinetic simulation

## Abstract

SHCS#879 is an ongoing Switzerland-wide multicenter observational study conducted within the Swiss HIV Cohort Study (SHCS) for the prospective follow-up of people living with HIV (PLWH) receiving long-acting injectable cabotegravir-rilpivirine (LAI-CAB/RPV). All adults under LAI-CAB/RPV and part of SHCS are enrolled in the project. The study addresses an integrated strategy of treatment monitoring outside the stringent frame of controlled clinical trials, based on relevant patient characteristics, clinical factors, potential drug-drug interactions, and measurement of circulating blood concentrations. So far, 91 blood samples from 46 PLWH have been collected. Most individuals are less than 50 years old, with relatively few comorbidities and comedications. The observed concentrations are globally in accordance with the available values reported in the randomized clinical trials. Yet, low RPV concentrations not exceeding twice the reported protein-adjusted 90% inhibitory concentration have been observed. Data available at present confirm a considerable between-patient variability overall. Based on the growing amount of PK data accumulated during this ongoing study, population pharmacokinetic analysis will characterize individual concentration-time profiles of LAI-CAB/RPV along with their variability in a real-life setting and their association with treatment response and tolerability, thus bringing key data for therapeutic monitoring and precision dosage adjustment of this novel long-acting therapy.

## 1. Introduction

Following the identification of HIV as the causal agent of AIDS in the early eighties, we have witnessed a progressive improvement in the management of HIV infection, which started with the approval of the first antiretroviral therapy (ART) for HIV treatment in 1987. Since the development of highly active ARTs about 25 years ago, complex therapies have been progressively simplified to potent, once-daily, fixed-dose multidrug formulations, thereby improving tolerability, efficacy, and convenience. These treatments have transformed HIV infection from a then fatal disease to a manageable chronic condition. Recently, not only the management but also the prevention of HIV infection has entered the bright new era of long-acting (LA) antiretroviral approaches, as reviewed elsewhere [1].

Cabotegravir (CAB) and rilpivirine (RPV), an integrase inhibitor combined with a non-nucleoside reverse transcriptase inhibitor, have been recently approved as a complete dual regimen for the maintenance treatment of HIV-1 infection in adults [2]. In Europe and Switzerland, CAB and RPV are marketed as two separate injectable medicines under the brand names VOCABRIA^®^ and REKAMBYS^®^, respectively, while CABENUVA^®^, a combined pack of CAB and RPV, is notably available in Canada and the United States. After an optional oral lead-in period of CAB 30 mg plus RPV 25 mg once daily, followed by an intramuscular (i.m.) loading dose (CAB/RPV at 600/900 mg), CAB and RPV are administered through i.m. injections either every 2 months (q8w) at 600/900 mg or monthly (q4w) at 400/600 mg. The nanosuspension technology enabled this long-acting injectable (LAI) delivery approach by increasing the apparent half-life of CAB and RPV from 41 h and 45 h to approximately 8.5 weeks and 20.5 weeks, respectively, although with substantial inter-individual variability already outlined in clinical trials [3]. Recently, LAI-CAB has also been approved by the U.S. FDA for use in at-risk adults and adolescents for pre-exposure prophylaxis (PrEP) to reduce the risk of sexually acquired HIV [4].

LA-ARTs undoubtedly have the potential to improve the treatment and prevention of HIV infection, particularly in terms of patient confidentiality, convenience, and empowerment. In addition, they will essentially contribute to overcoming the challenge of adherence. However, we hypothesize that close monitoring of the patients will probably be necessary for an optimal implementation of these revolutionary approaches. Indeed, many people living with HIV (PLWH) face complex situations, which are rarely taken into account in most clinical trials [5,6,7,8]. Concomitant initiation of treatments for comorbidities with a definite risk of drug–drug interactions (DDIs) (e.g., tuberculosis, HCV infection, cancer, and further comorbidities associated with polypharmacy) will definitely require special attention in people receiving this new LA-ART. Moreover, LAI-CAB/RPV therapy will require close monitoring in underweight or obese people and possibly in pregnant women with regard to not only efficacy but also tolerability and long-term safety. In particular, a body mass index (BMI) greater than 30 already appears to be an independent risk factor for CAB/RPV treatment failure [9].

In this context, we have launched a Swiss-wide prospective observational study within the frame of the Swiss HIV Cohort Study (SHCS) (project SHCS#879). This project aims to bring an original contribution to the monitoring of the LAI-ART by investigating the characteristics of the LAI-CAB/RPV pharmacokinetics (PK) in real-life PLWH. Despite the so far limited clinical validation of therapeutic drug monitoring (TDM) in the context of ART [10], we believe that TDM will be an important component of optimal patient follow-up in the LA-ART era. Indeed, with adherence no longer representing a confounding factor (as long as the patients do not miss their injections’ appointments), physicians facing inadequate therapeutic response in patients are likely to question whether they are actually exposed to appropriate levels of antiretroviral drugs throughout the whole dosing intervals. Although population pharmacokinetic (popPK) analyses of CAB/RPV have been conducted during phase III clinical trials [11,12], there is, to the best of our knowledge, no popPK analysis involving a real-life cohort of PLWH. The SHCS#879 observational study addresses an integrated strategy of LAI-ART treatment monitoring outside the stringent frame of controlled clinical trials, based on relevant individual characteristics, demographic/clinical factors, potential DDIs, and measurement of circulating blood concentrations. Based on the information continuously gathered during the ongoing implementation of LAI-CAB/RPV within the Swiss PLWH population, a popPK analysis will be performed to characterize the variability of concentration-time profiles of LAI-CAB/RPV and to examine the patients’ covariates possibly influencing LAI-ART plasma exposure. Ultimately, the SHCS#879 project aims to develop a suitable popPK model, providing an individual prediction of the range of trough concentration (C_min_) expected in a given PLWH, taking into account both the known variability of drug concentrations and the patient’s individual characteristics. The present article gives some insights into the first observations performed during the implementation of LAI-CAB/RPV in Switzerland and examines whether they fit with our underlying hypothesis on the clinical relevance of TDM.

## 2. Materials and Methods

### 2.1. Study Design

The SHCS, established in 1988, is an ongoing multicenter, clinic-based, prospective, longitudinal, observational study including HIV-infected adults in Switzerland [13,14]. SHCS#879 is an ongoing Swiss-wide prospective observational project on LAI-CAB/RPV in a real-life cohort of PLWH. Adults (>18 years old) enrolled in the SHCS and followed up in the centers of Lausanne, Geneva, Bern, Basel, Zürich, St-Gall, and Lugano are systematically included in the SHCS#879 project if they receive the CAB-RPV regimen. At present, physicians are selecting in the first instance, PLWH who have been consistently adherent to antiretroviral therapy, although this is not an inclusion criterion, in order to avoid potential issues related to administration schedules (e.g., patients who forget or fail to show up for drug administration appointments). In addition, a one-month oral lead-in period is currently recommended in Switzerland [15,16], but its implementation occurs according to physician and patient preferences. For the most part, the samples are collected and the data are recorded at medical visits, on average every 2 months, or at one or two of the bi-yearly planned SHCS cohort visits. These cohort visits do not necessarily coincide with the day of CAB/RPV injection, resulting in samples obtained at unselected times during the entire dosing interval. In addition, to enrich the data collected during standard visits, detailed PK investigations are planned in PLWH who consent to donate a few additional blood samples. These additional investigations will allow a better characterization of the PK of early levels after i.m. injection and capture the actual C_min_ concentrations (just prior to next i.m. injection). Furthermore, considering the reported existence of secondary depots of drug nanoparticles distributed throughout the lymphatic and reticuloendothelial systems [17], CAB and RPV levels will be quantified in peripheral blood mononuclear cells (PBMC) from a subset of consenting patients.

The study data (i.e., demographic parameters, adverse events, CAB/RPV dosing regimen) are recorded using the study TDM request form, complemented with relevant clinical information (i.e., CD4 count, viremia, lab values, comedications) extracted from the national SHCS database, which contains the prospective medical records of SHCS cohort visits. In particular, the date and time of both the blood sampling and injection of LAI-CAB/RPV are carefully documented in the TDM request form, as the time lapse after the dose is necessary to interpret plasma concentrations.

Of note, some individuals were receiving CAB and RPV prior to Swiss market authorization and the start of the SHCS#879 study. Some samples were collected from PLWH receiving treatment for compassionate use, while other samples were obtained as laboratory quality control within the frame of the analytical method development. All these data could be included in the study since these individuals are still followed in their respective SHCS centers.

### 2.2. Outcomes

The primary study endpoint is the quantification of CAB/RPV concentrations in plasma, and in PBMC, for the development of detailed popPK models and the characterization of the secondary depots of drug nanoparticles in PBMC, respectively. The key secondary endpoint is the quantification of CAB/RPV C_min_ in plasma as part of a validation step assessing whether the PK model is able to predict actual C_min_ based on one random sample drawn at an unselected time during the dosing interval. Other secondary endpoints include the comparison of PK characteristics from real-life patients on LAI-ART with those deduced from clinical trials submitted to registration agencies, the association of “on-target” plasma levels with treatment efficacy and tolerability, the assessment of the proportion of patients who interrupt i.m. treatment and the subsequent analysis of predictors for the interruption, and the identification of factors leading to a viral failure (defined as confirmed HIV viral load > 200 copies/mL). Targeted and detailed exploration of specific clinical situations is planned as well, such as undescribed DDIs or impact of specific comorbidities, susceptible to affect the absorption and disposition of LAI-ART. None of these endpoints of the SHCS#879 study have yet been attained at this early stage of the study, and thus, this article focuses only on preliminary results indicating whether or not they support our underlying assumption on the clinical interest of TDM.

### 2.3. Drug Level Measurements

The Laboratory of Clinical Pharmacology (CHUV, Lausanne, Switzerland) is equipped with a state-of-the-art platform of six instruments of high-performance chromatography coupled to tandem mass spectrometry (LC-MS/MS). It participates in International External Quality Control Proficiency Programs for CAB and RPV (Asqualab, Paris, France, https://www.asqualab.com/ accessed on 26 May 2022; KKGT, The Hague, The Netherlands, http://kkgt.nl/?lang=en accessed on 26 May 2022), certifying accurate drug level measurements. Whole blood samples are shipped immediately in a plastic transport protection coffer to the Laboratory of Clinical Pharmacology. In-house stability studies indicate that CAB and RPV are stable in the whole blood at room temperature for up to 96 h. Upon arrival, blood samples are centrifuged (2000× *g*, 10 min at 4 °C), and the separated plasma is frozen at −80 °C until analysis by LC-MS/MS, using a previously published validated multiplex method [18]. The CAB and RPV’s lower limits of quantification are 25 ng/mL and 5 ng/mL, respectively. An analytical series for LC-MS/MS quantification is performed once a week, and the TDM results are sent within the same week to the physicians in charge.

Concerning the investigations on PBMC, the quantification of cell-associated ART levels require cell isolation immediately after blood collection using Vacutainer^TM^ Cell Preparation Tubes (CPT), using an adaptation of methods developed in our laboratory [19,20]. Cellular isolation is conducted according to the standardized procedure provided by the manufacturer [21].

### 2.4. Pharmacokinetic Modelling

Circulating drug concentration in plasma is the principal driver for both the efficacy and toxicity of most systemic therapies. Yet, drugs are often prescribed at standard dosages without taking into account between-patient PK variability, which can be remarkably large for some HIV drugs with a definite impact on therapeutic response. Multiple sources of variability have been identified, including demographic, environmental, clinical, pharmacological, and genetic factors. Population-based approaches represent the best way to characterize the PK profile of drugs in a cohort of patients and to capture the contribution of individual factors affecting drug levels [22,23].

The characterization of the LAI-CAB/RPV PK in PLWH enrolled in this study will be performed using compartmental methods employing non-linear mixed effects modeling techniques. We will develop popPK models for CAB and RPV to describe their absorption and disposition after the injection of LA formulations and determine the effects of specific factors such as sex, age, body size, comorbidities, DDIs, and issues at the injection site or concomitant pathophysiological conditions. These models will be used to derive Bayesian maximum likelihood indicators of drug exposure (C_min_ levels). As most (up to 97%) of the SHCS participants have given their consent for genetic exploration, the popPK model developed for the SHCS#879 study will also be used for pharmacogenetic explorations.

At this early stage of results examination, our essential question was simply whether these first concentration data of LAI-CAB/RPV supported our underlying hypothesis of a clinical interest in TDM. The first criterion for TDM candidates is a significant between-subject PK variability associated with acceptable within-subject PK stability over time [24]. For our preliminary concentration data, this translates into statistically comparing the inter-individual variability and the within-patient variability with a one-way analysis of variance on log-transformed values, neglecting to a first approximation the influence of time lapse after dose and various covariates.

## 3. Preliminary Results

### 3.1. Study Population

At this early stage of the ongoing study (last samples collected in July 2022), 46 PLWH from Lausanne, Zürich, Bern, Geneva, and Basel have been so far included in SHCS#879. Table 1 summarizes their demographic and clinical characteristics.

To date, most of the included individuals are less than 50 years old (61%, 28 PLWH), male (83%, 38 PLWH), and Caucasian (63%, 29 PLWH), with relatively few comorbidities. In addition, seven PLWH (15%) are considered obese (BMI higher than 30 kg/m^2^), while more than half (54%, 25 PLWH) are classified as overweight (BMI between 25 and 30 kg/m^2^) [27]. No virologic failure, defined as confirmed HIV viral load > 200 copies/mL, occurred, but viral blips (single HIV viral load < 200 copies/mL) were observed in three patients (7%). Overall, 94% of the PLWH were undetectable (HIV viral load < 50 copies/mL), and one had missing data.

Individuals who were receiving CAB/RPV for compassionate use prior to Swiss market authorization received LAI-CAB/RPV every 4 weeks (4%, 2 PLWH), as this was the only recommended regimen at that time. However, these PLWH are to switch to the q8w regimen, with one individual having switched very recently. Participants who started the LAI treatment during the SHCS#879 study all receive the q8w regimen (96%, 44 PLWH). The oral lead-in was not initiated for every PLWH, with 17% having started directly with the i.m. loading dose. Lastly, various comedications were recorded, with five PLWH (30%) having three or more comedications. None were considered likely to cause DDIs.

### 3.2. Adverse Events

Table 2 presents the main categories of reported adverse events.

The most common recorded adverse events were injection-site reactions (12%), reported in eight PLWH (17%). Most people did not report any side effects (70%, 32 PLWH), while seven individuals (15%) reported multiple adverse events.

### 3.3. Pharmacokinetics

Overall, 91 samples, of which 61 were collected during the long-acting dosing interval, were obtained from the 46 PLWH so far enrolled in the study. Thirty samples were obtained during or at the end of the oral lead-in period. Figure 1 shows the concentrations observed up to now, compared to usual ranges (approximated for illustrative purposes) reported in clinical trials [8,28,29,30,31].

Observed concentrations confirm the hypothesis of an inter-individual variability much larger than the within-patient variability for both CAB (101% vs. 50%, *p* = 0.002) and RPV (94% vs. 27%, *p* < 0.0001). The concentrations observed are globally in accordance with the ranges already reported in the literature. However, low RPV concentrations at approximately two times the PAIC_90_ were observed in six individuals (13%). One patient (receiving CAB/RPV q8w, in orange) showed a surprisingly rapid concentration decrease during the follow-up injection, without identified cause. This patient’s medical records containing historic TDM results performed during its previous oral therapy indicated adequate exposure to bictegravir, also a substrate of CYP3A and UGT1A1, as are rilpivirine and cabotegravir, respectively [34]. On the other hand, this individual is young and athletic and was injecting anabolic steroids before starting LAI-CAB/RPV treatment, which might suggest undisclosed exposure to an interacting treatment. Physical activity could also result in greater absorption and, therefore, faster elimination [3]. Overall, the majority of the blood samples were collected at or close to C_min_ (85%, 77 samples). Notably, seven samples (8%) were collected between 9 and 10 weeks after the last drug administration.

## 4. Discussion

The population included so far in the SHCS#879 project is relatively healthy and, therefore, similar to the population observed in clinical trials. However, significant PK variability is already observed, which confirms an important prerequisite for the rational development of a TDM strategy. In particular, some RPV levels only reached about two times the reported PAIC_90_ of 12 ng/mL. Such low levels of circulating RPV could be of concern in terms of efficacy and safety. In clinical practice, a minimum plasma concentration of 50 ng/mL is sometimes recommended [35,36]. Some authors even conclude that higher plasma levels should be targeted [37]. In our study, one individual had not only low levels of RPV but also low CAB for several weeks and showed a surprisingly rapid decrease over one injection interval. Various hypotheses were considered, and a preliminary investigation revealed that this individual was presumably not a rapid metabolizer of CYP3A and/or UGT1A1, which could have explained the abnormally low plasma drug levels. Despite the lack of definite clinical evidence [38,39,40], long-lasting physiologic or inductive effects resulting from previous or reiterated use of anabolic steroids, such as an increased intrinsic organ clearance or increased metabolic activity may last for weeks to months, and could explain, to some extent at least, the observations. In addition, depending on the type of steroid-conjugated fatty acid used, a slower release from the injected depot could also have contributed to sustained physiologic or metabolic effects [38]. Eventually, nearly a year after the start of the treatment, RPV and CAB concentrations reached levels within the reported ranges. Although the timing may be quite revealing, further investigation is needed to clearly establish the causal relationship. This case nevertheless highlights the potential importance of TDM to prevent under- or over-exposure that could lead to therapeutic failure or toxicity, respectively. In addition to the standard monitoring of viral suppression and CD4 counts, TDM might become advised for ensuring optimal therapeutic efficacy. Note that due to the small number of patients included at this time, it is difficult to determine whether lower drug levels could cause viral blips.

The initial stage of the implementation of LAI-CAB/RPV in Switzerland has already provided clinically useful information for improved patient management. In particular, at the Lausanne center, the oral lead-in period has been reduced to only 3 weeks as the oral steady-state is largely achieved after 21 days, and drug tolerability is certainly assessed. The remnant tablets spared from week 4 are then available to complete the regimen in case patients would miss or are unable to attend a scheduled i.m. appointment. On another note, it appears that CAB injection is less painful than RPV injection. It is, therefore, preferable to inject CAB first for the comfort of the patient. This recommendation is not included in the official monographs but could have an impact on the long-term acceptability of these treatments. Indeed, one of the main limitations associated with these drugs is the large volumes injected, which can be painful and for which there is currently no alternative. Various formulations are currently being developed to address this issue and may eventually allow for longer administration intervals [41,42,43].

On another note, the majority of samples were, expectedly, collected at or close to C_min_, usually at the end of the optional oral lead-in period, just prior to the loading dose, and then one month later, just prior to the first injection for the maintenance regimen. It is indeed more convenient for healthcare professionals to perform blood sampling just before the next injection. The patient is then asked to come only on the days when the drugs are to be administered, i.e., every 2 months. It might be convenient sometimes, however, to collect TDM samples at random times during the injection interval. When sufficient data are available, popPK models will be developed for CAB and RPV using parametric non-linear mixed effect modeling and validated by comparing the inferred PK-models predictions with actual LAI-ART C_min_ measured in patients. This clinical validation will also indicate whether an intervention based on TDM might improve antivirals plasma exposure in patients on LAI-ART, thereby giving indications on the potential suitability of altering the LAI-ART dosing schedule by shortening or extending dosing intervals in selected patients exhibiting lower or higher C_min_, respectively. Considering the timely implication of LAI-ARTs in the management of HIV, the development of popPK models will help improve the implementation of a clinically-appropriate TDM for these new drugs in real-life situations.

As these are preliminary results from the implementation of LAI-CAB/RPV in Switzerland, only limited data are available at this time. As described above, physicians are at present selecting PLWH who have been consistently adherent to antiretroviral therapy. In addition, the organization, staff, and infrastructure are currently evolving to accommodate the new paradigm represented by these novel approaches. These issues limit, to some extent, the ability to enroll patients at this time. We ultimately aim at including 200–300 PLWH over the next two years, with at least five samples collected per person, thus reaching a total of more than 1000 blood samples. Our translational research collaboration encompasses state-of-the-art mass spectrometry assays, access to institutional genetic platforms, prospective capture of PK, pharmacodynamic, genetic, metabolic markers and clinical data, modeling and simulation capabilities, and clinical expertise in TDM. This setting, therefore, offers a unique opportunity for contributing to the short- and long-term optimization of LAI-ARTs at the individual patient level.

## Figures and Tables

**Figure 1 pharmaceutics-14-01588-f001:**
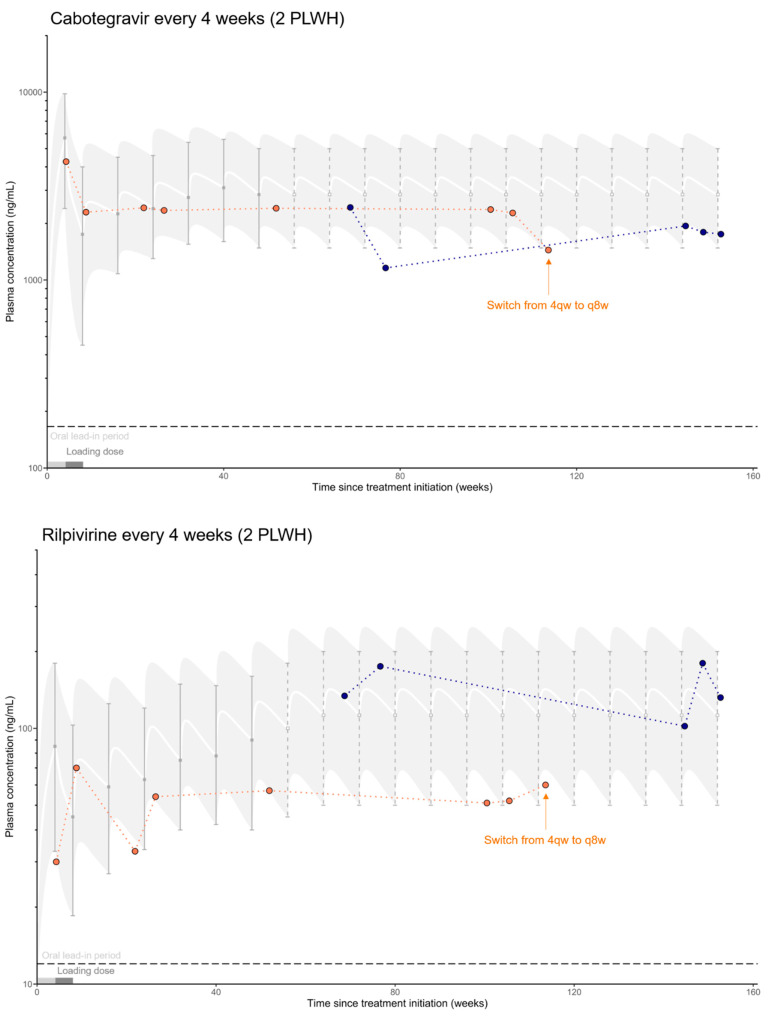
Observed plasma concentrations for the different dosing regimen of CAB and RPV, along with usual concentration ranges, approximated for illustrative purposes, from the ATLAS-2M trial shown as median profile and 5% and 95% percentiles [8]. Ranges are prolonged beyond 48 weeks for CAB as PK steady-state is reached after 44 weeks (dashed intervals) [29]. For RPV, 80% of the PK steady-state is achieved after 52 weeks [30], so the data were extrapolated accordingly, and showed accordance with the results of the FLAIR trial [28]. For visual purposes, dotted lines connect drug levels measurements (different color of dots for each PLWH) in the same PLWH. The horizontal dashed lines represent the protein-adjusted inhibitory concentration required for 90% inhibition (PAIC_90_). PAIC_90_ values are 166 ng/mL for CAB [32], and 12 ng/mL for RPV [33]. The oral lead-in period followed by the loading dose period are indicated, prior to the initiation of maintenance therapy. Finally, because concentrations from ATLAS-2M were obtained in patients who underwent a 4-week oral lead-in period, data from individuals included in this present study who started directly with injections were shifted accordingly to match the *x*-axis. PLWH: people living with HIV; q4w: every 4 weeks; q8w: every 8 weeks.

**Table 1 pharmaceutics-14-01588-t001:** Characteristics of the PLWH included in the SHCS#879 study.

Population Characteristics Recorded at Last Cohort Visit or Last Sample Collection (n = 46)	Median (Range) or Number (%)	[Missing Data, (%)]
**Demographic characteristics**		
Sex:		
Male	38 (83)	
Female	8 (17)	
Ethnicity:		
Caucasian	29 (63)	
Black	6 (13)	
Hispanic American	3 (7)	
Asian	3 (7)	
Unknown	5 (10)	
Age (year)	45 (28–62)	
Body weight (kg)	83 (63–120)	
Height (cm)	177 (161–189)	
BMI (kg/m^2^)	26 (19–37)	
**Physiological characteristics**		
Serum creatinine (μmol/L)	85 (46–131)	[4, (9)]
CL_CR_ (mL/min) ^a^	111 (61–176)	[4, (9)]
eGFR (mL/min/1.73 m^2^) ^b^	92 (44–145)	[4, (9)]
CKD stage (mL/min/1.73 m^2^)		
G1: ≥90	21 (46)	[4, (9)]
G2: 60–89	19 (41)	
G3a: 45–59	1 (2)	
G3b: 30–44	1 (2)	
Liver function		
Albumin (g/L)	44 (36–51)	[17, (37)]
Bilirubin (μmol/L)	7 (3–19)	[13, (28)]
AST (UI/L)	24 (13–44)	[4, (9)]
ALT (UI/L)	26 (9–67)	[4, (9)]
Heart blood pressure:		
Diastolic pressure (mmHg)	80 (60–107)	[4, (9)]
Systolic pressure (mmHg)	130 (108–180)	[4, (9)]
Malabsorption after gastrectomy	1 (2)	
**HIV molecular biology**		
CD4 (cells/mm^3^)	667 (191–1192)	[4, (9)]
HIV RNA (copies/mL)		
<50	43 (94)	[1, (2)]
>50 and <200	2 (4)	
**Previous antiretroviral therapy, no. (%):**		
Bictegravir/Tenofovir alafenamide/Emtricitabine	18 (39)	[5, (10)]
Elvitegravir/Tenofovir alafenamide/Emtricitabine/Cobicistat	6 (13)	
Dolutegravir/Lamivudine	3 (7)	
Darunavir/Tenofovir alafenamide/Emtricitabine/Cobicistat	3 (7)	
Other	11 (24)	
**Antiretroviral therapy, no. (%):**		
Long-acting regimen		
CAB-RPV q4w	2 (4)	
CAB-RPV q8w	44 (96)	
Followed oral lead-in period	38 (83)	
**Number of comedications, no. (%):**		
0	15 (33)	[4, (9)]
1	10 (22)	
2	3 (7)	
3	5 (10)	
4	1 (2)	
≥5	8 (17)	

CKD: Chronic Kidney Disease; CL_CR_: Creatinine Clearance; eGFR: estimated Glomerular Filtration Rate; AST: aspartate aminotransferase; ALT: alanine aminotransferase; q4w: every 4 weeks; q8w: every 8 weeks. ^a^ CL_CR_ calculated according to the Cockcroft and Gault equation [25]. ^b^ eGFR calculated according to the CKD-EPI equations reported by Levey et al. [26]. Note that some PLWH were recently enrolled in the SHCS, while four PLWH are in the SHCS process enrolment. Therefore, much of the data for these individuals are not yet available at this time.

**Table 2 pharmaceutics-14-01588-t002:** Main categories of reported adverse events.

Adverse Events Categories	Number Reported in the TDM Request Forms (%) (n = 91)	Number of PLWH (%) (n = 46)
No adverse events, no. (%)	67 (74%)	32 (70%)
Any adverse event, no. (%)	20 (22%)	14 (30%)
Injection site reaction ^a^	11 (12%)	8 (17%)
Pyrexia ^b^	1 (1%)	1 (2%)
Fatigue ^c^	5 (5%)	4 (9%)
Headache	3 (3%)	2 (4%)
Musculoskeletal pain ^d^	5 (5%)	4 (9%)
Gastrointestinal disorders ^e^	2 (2%)	2 (4%)
Sleep disorders ^f^	2 (2%)	2 (4%)
Missing data	4 (4%)	-

^a^ Includes pain/discomfort, nodules, induration, swelling, erythema, pruritis, bruising, discoloration, warmth, hematoma. ^b^ Includes pyrexia, feeling hot, chills, influenza-like illness, body temperature increased. ^c^ Includes fatigue, malaise, asthenia. ^d^ Includes musculoskeletal pain, musculoskeletal discomfort, back pain, myalgia, pain in extremity. ^e^ Includes nausea, dizziness, diarrhea.^f^ Includes insomnia, poor quality sleep, somnolence.

## Data Availability

The data presented in this study are available on request from the corresponding author.

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
