# Peer review of "Real-Life Therapeutic Concentration Monitoring of Long-Acting Cabotegravir and Rilpivirine: Preliminary Results of an Ongoing Prospective Observational Study in Switzerland"

_pharmaceutics, 2022, doi:10.3390/pharmaceutics14081588_

Round 1

Reviewer 1 Report

The work is significant and effectively addresses the problems raised in the title. I believe it will be beneficial if it is published and distributed to the scientific community. In fact, the text needed considerable grammar and English language correction. I submitted the paper for plagiarism detection, and I am accepting it with minimal changes in the hopes that it will not be rejected.

Author Response

We thank the reviewer for the positive global appreciation of our article and our study.

We have checked English language and style. We hope that the reviewer will find the revised version suitable for publication.

Reviewer 2 Report

The present study by Thoueille et al is highly significant. With long-acting ART (LA-ART) therapy, we anticipate many more such kinds of studies. Moreover, for the success of LA-ART therapy continuously performing this kind of analysis are highly crucial and essential. 

Author Response

We thank the reviewer for the very favorable appreciation of our article and enthusiastic comments towards our study.

The English language and style have been checked.

Reviewer 3 Report

The study brings important info on real-life data of long-acting cabotegravir and rilpivirine.

Author Response

We certainly acknowledge that the article may be improved.

As stated in the article, this is an ongoing study, and a substantial number of additional patients and new PK data has been available since the time of article submission: we went from 15 patients with 39 samples collected, to 46 patients with 91 samples collected. These additional data have been incorporated into the updated analyses. The consolidated results strengthen our preliminary observations on the long-acting cabotegravir and rilpivirine in a real-life cohort of people living with HIV

According to the recommendations, the English language and style have been checked.